# Forest, Crop and Grassland Leaf Area Index Estimation Using Remote Sensing: A Review of Current Research Methods, Sensors, Estimation Models and Accomplishments

Nokukhanya Mthembu [1], Romano Lottering [2,*] and Heyns Kotze [1]

1   Forest Operations, Mondi House, 380 Old Howick Road, Pietermaritzburg 3200, South Africa
2   Discipline of Geography, School of Agricultural, Earth and Environmental Sciences, University of KwaZulu-Natal, Pietermaritzburg 3209, South Africa
*   Correspondence: lottering@ukzn.ac.za

**Abstract:** Leaf area index (LAI) is an important parameter in plant ecophysiology; it can be used to quantify foliage directly and as a measure of the photosynthetic active area and, thus, the area subject to transpiration in vegetation. The aim of this paper was to review work on remote sensing methods of estimating LAI across different forest ecosystems, crops and grasslands in terms of remote sensing platforms, sensors and models. To achieve this aim, scholarly articles with the title or keywords "Leaf Area Index estimation" or "LAI estimation" were searched on Google Scholar and Web of Science with a date range between 2010 and 2020. The study's results revealed that during the last decade, the use of remote sensing to estimate and map LAI increased for crops and natural forests. However, there is still a need for more research concerning commercial forests and grasslands, as the number of studies remains low. Of the 84 studies related to forests, 60 were related to natural forests and 24 were related to commercial forests. In terms of model types, empirical models were most often used for estimating the LAI of forests, followed by physical models.

**Keywords:** LAI estimation; sensors; models; passive; active; data source LAI products

## 1. Introduction

Leaf area index (LAI) is an important parameter in plant ecophysiology, as it can be used to quantify foliage directly and as a measure of the photosynthetic active area; thus, the area is subjected to transpiration in vegetation [1,2]. The importance of LAI is seen in applications ranging from process-based ecosystem simulations to site water balance and radiative transfer studies. Furthermore, LAI is a major input variable for Physiological Processes Predicting Growth (3PGS) models to predict the growth of different species and water use [3]. LAI estimations and mapping of forests, crops and grasslands date back to the 1990s. Recent studies of forest, crop and grassland LAIs were published by [4–7].

Many definitions of LAI were proposed in the literature, and these definitions were often dependent on the purpose of the study. For the purpose of this study, the following definition will be used: LAI is the total one-sided area of leaf tissue per unit ground surface area [1]. According to this definition, LAI quantitatively measures the leaf surface area available for the interception of photosynthetically active radiation (PAR) and transpiration and is an important structural variable for describing the energy and mass exchange in a vegetated ecosystem [8].

There are various methods of measuring LAI, which can be broadly grouped into two categories: direct and indirect methods. Direct methods of estimating LAI include harvesting, litter collection and allometry and are more accurate than indirect methods [9]. Indirect methods include optical devices such as LICOR-2200, which measures intercepted radiation below the vegetation canopy [10]. Deriving LAI from remote sensing data is another indirect method of estimating LAI. Both these methods of estimating LAI offer

different advantages and disadvantages. However, indirect methods were shown to offer more advantages than disadvantages, especially when a large area was studied. Remote sensing is one of the commonly used indirect methods of estimating LAI, as remote sensing technology offers a better alternative to estimating and mapping LAI for larger landscapes more efficiently and accurately [11]. Remote sensing also offers a less time-consuming and cost-effective method of estimating LAI [12].

The development of high spatial resolution satellite data enabled researchers around the world to more effectively monitor vegetation at higher accuracy [13]. Remote sensing enables convenient collection of data dating back to several years while providing reliable and accurate estimates of LAI and other biophysical attributes for different vegetation types. Although a number of remote sensing methods of estimating LAI were developed, not a single method can be applied consistently and repeatedly for estimating LAI locally and regionally [14]. The reason for this is variations in biophysical, environmental and topographic traits of vegetation in space and time [15,16].

LAI can be derived by using empirical, statistical and hybrid methods. Deriving LAI by means of statistical and physical approaches was first carried out on crop canopies in the beginning of the 1970s. One of the earliest studies on estimating LAI using statistical approaches was carried out in 1974 by [17], who used Landsat MSS to derive wheat LAI. In recent years, the number of studies on crop LAI increased, while recent studies on crop LAI were published by [9,13,18,19]. After research on crop LAI estimations using statistical approaches started showing positive results in the 1980s, the number of forest LAI studies increased. With regard to commercial forests and grasslands, there is still a limited number of studies; therefore, this area requires further investigation.

During the last decade, the use of remote sensing to estimate and map LAI increased [20]. High spatial and spectral resolution sensors such as QuickBird, IKONOS and WorldView-3 showed great potential in achieving acceptable levels of accuracy in estimating LAI. The availability of high spatial resolution sensors, such as the ones mentioned above, also led researchers to investigate the effects of spatial and spectral resolution in effectively estimating LAI. A number of studies were conducted about satellite remote sensing for forestry planning to show that 30 m spatial resolution provides insignificant results when it comes to forest management planning. Ref. [21] conducted a similar study as the one mentioned above but used a higher spatial resolution satellite sensor. In this study, forest structural attributes such as diameter at breast height (DBH), mean tree height (MTH) volume and basal area (BA) were investigated for the same species and in the same geographical area but results were different. In terms of coefficient of determination, results were as high as 0.64 for DBH.

Advancements in remote sensing made it possible to estimate LAI timely with acceptable accuracy. A number of previous studies estimated LAI with acceptable accuracies using spectral reflectance; however, due to saturation of vegetation indices at LAI values above three, texture measures were utilized as an alternative (Gu et al., 2013). Image texture is the measure of the spatial variation in the grey levels in the image as a function of scale. Image texture was demonstrated by Gu et al. (2013), Pu and Cheng (2015) and Li et al. (2019) to improve the estimation of LAI.

This paper reviews previous work on remote sensing methods of estimating LAI across different forest ecosystems, crops and grasslands. These three landcover types are distinct in characteristics and, therefore, are studied using different methods, equipment and models. Many different types of forests exist but they are generally characterized by different tree species with distinct upper and lower canopy layers. These forests can be either planted or grow naturally. The different types of forests include tropical, temperate, boreal and savanna forests. Crops are a cultivated type of vegetation that are planted on a large scale. Crops are grown and harvested for profits or subsistence. Examples of crops include but are not limited to rice, wheat and maize [9]. A grassland is a large open area where vegetation is dominated by grasses. Grasslands generally grow naturally and are

found in most regions of the world. Grasslands and crops are relatively shorter plants compared to most types of forests [18].

This review investigates the current trends and accomplishments and direction of research in terms of different remote sensing platforms, sensors and models that are utilized to estimate LAI. To achieve this, the paper reviewed published papers on LAIs estimated using remote sensing from the past ten years. Remote sensing was used by a number of researchers to estimate LAI globally and the availability of different remote sensing imagery and increases in diversity of spatial, spectral, temporal and radiometric characteristics led to more research and, thus, better accuracy in estimating LAI [1]. There was a sharp increase in the number of LAI studies using remote sensing between 2010 and 2020; this review, therefore, investigated the contributing factors to this increase, which were also reported in the results of a review paper by Xu et al. [12].

## 2. Materials and Methods

Scholarly articles with the title or key words "Leaf Area Index estimation" or "LAI estimation" were searched on Google Scholar and Web of Science with a date range between 2010 and 2020. In total, 490 articles were found, 312 were found from Google Scholar and 178 were found on the Web of Science. These articles were downloaded into Endnote (The EndNote Team, 2013, Philadelphia, PA, USA), and thereafter, duplicated articles were discarded. The remaining articles were then filtered to eliminate conference papers, review papers, theses, book chapters and audio-visual material. Studies focusing on LAI estimations using ground-based handheld devices and studies focusing on LAI estimations for vegetation cover types of no interest were also removed. This reduced the dataset to 168 journal articles from 23 different journals (Table 1). These journal articles were then manually screened to study sites, satellite sensor used and statistical method used to determine LAI. Of the 168 papers reviewed, 157 used optical devices to estimate LAI data and only 8 papers measured LAI using ground-based methods.

**Table 1.** Journal list and the number of articles for forest, crop and grassland.

| Journal | Number of Publications |
| --- | --- |
| Remote Sensing of Environment | 38 |
| Remote Sensing | 26 |
| International Journal of Remote Sensing | 18 |
| IEEE Transactions on Geoscience and Remote Sensing | 10 |
| International Journal of Applied Earth Observation and Geoinformation | 9 |
| Agricultural and Forest Meteorology | 9 |
| Canadian Journal of Remote Sensing | 9 |
| ISPRS Journal of Photogrammetry and Remote Sensing | 6 |
| IEEE Journal of Selected Topics in Applied Earth Observations and Remote Sensing | 6 |
| Forests | 6 |
| GIScience & Remote Sensing | 5 |
| Remote Sensing Letters | 4 |
| Sensors | 4 |
| Journal of Forestry Research | 4 |
| Forest Ecology and Management | 3 |

**Table 1.** *Cont.*

| Journal | Number of Publications |
| --- | --- |
| European Journal of Agronomy | 2 |
| Chinese Journal of Geophysics | 2 |
| Journal of Quantitative Spectroscopy and Radiative Transfer | 2 |
| Annals of Forest Science | 1 |
| Precision Agriculture | 1 |
| Plant Methods | 1 |
| Journal of Geography, Environment and Earth Science International | 1 |
| Journal of Remote Sensing | 1 |

## 3. Results

A total of 168 journal articles were screened from 23 different journals (Table 1). Remote Sensing of Environment had the most publications (38), followed by Remote Sensing (26), International Journal of Remote Sensing (18) and IEEE Transactions on Geoscience and Remote Sensing (10). Journals such as International Journal of Applied Earth Observation and Geoinformation, Agricultural and Forest Meteorology and Canadian Journal of Remote Sensing had less than 10 journal articles, while journals such as Journal of Geography, Environment and Earth Science International, Precision Agriculture and Plant Methods had one publication each.

### 3.1. Application Areas

As mentioned above, this paper reviewed 168 papers related to estimate LAI across different forest ecosystems, crops and grasslands. Forests had the highest number of studies, i.e., 84 studies, followed by crops, with 56 studies, and grasslands, with 25 studies (Figure 1). Forest ecosystems were broadly categorized into two groups: natural forests and commercial forests, and of the 84 studies related to forests, 60 studies were related to natural studies and 24 studies were related to commercial forests (Figure 2).

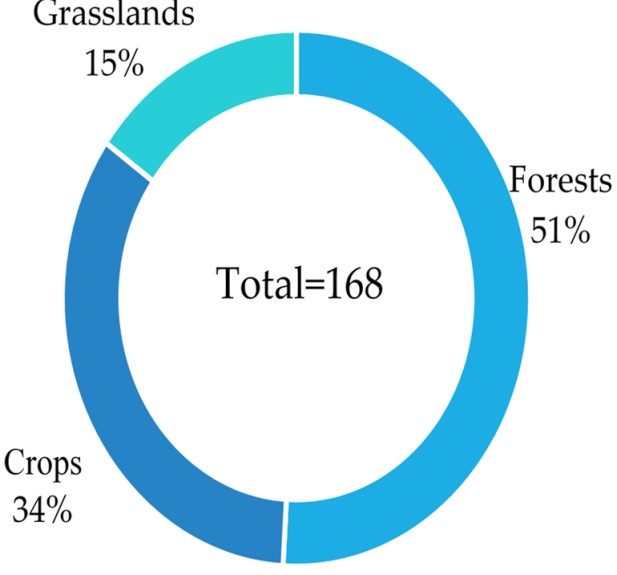

**Figure 1.** Number of LAI studies for forests, crops and grasslands.

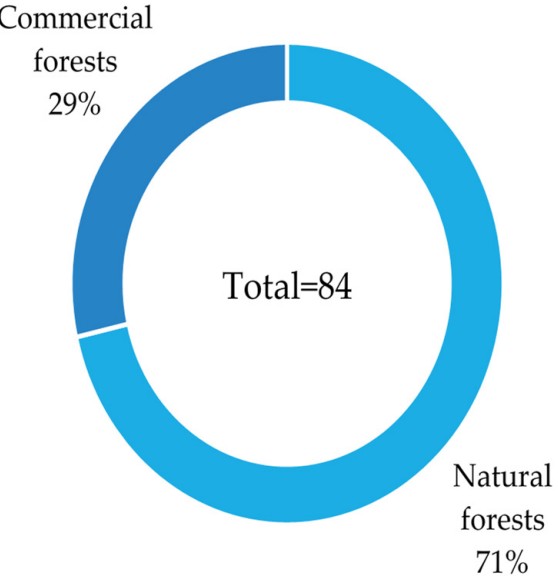

**Figure 2.** Number of LAI studies for natural forests and commercial forests.

### 3.2. Spatial Distribution of LAI Studies

Of the 168 studies that were reviewed, 12% were conducted at a global level. At the continental level, North America (12) had the highest number of studies, followed by South America (9), Europe (6), Asia (5), Australia (4) and Africa (3) (Figure 3). Fourteen studies were conducted at variable levels.

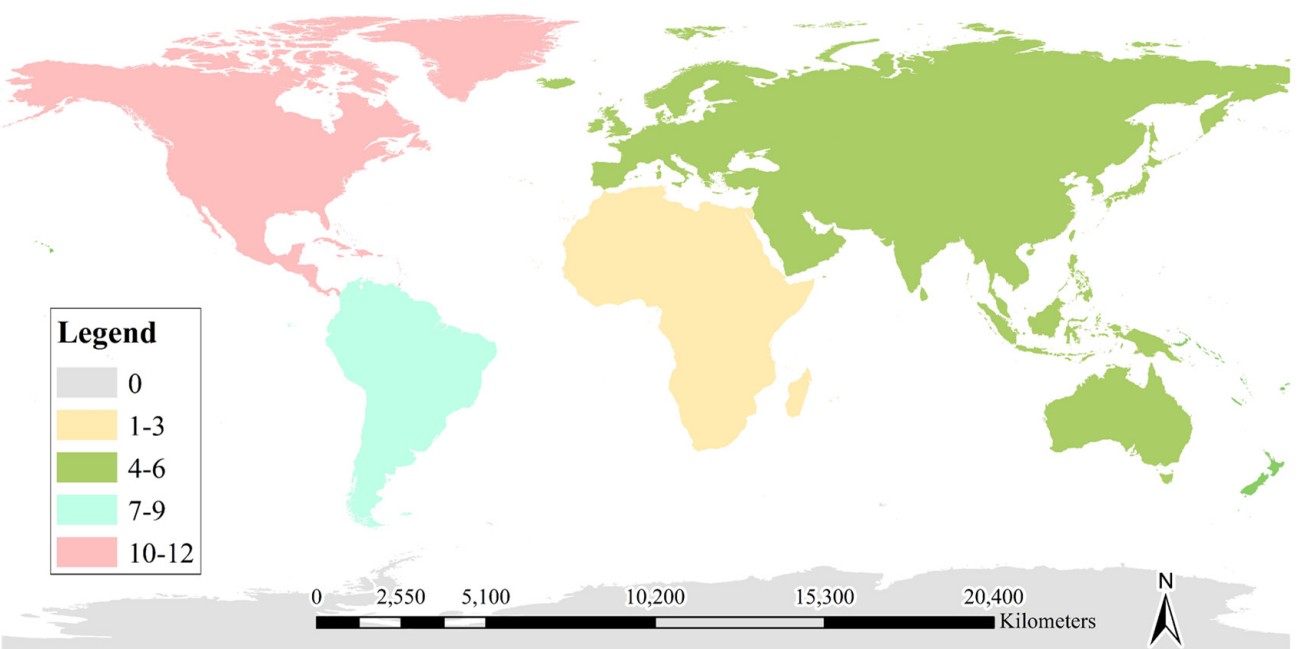

**Figure 3.** Spatial distribution and number of studies conducted at continental level.

The spatial distribution of studies at country and sub country scale showed that China (29) had the highest number of studies, followed by the US (21), Brazil (16), Australia and Spain (11) and New Zealand (10) (Figure 4). There were a number of countries in Africa, Asia, South America and the Middle East that had no studies at country scale.

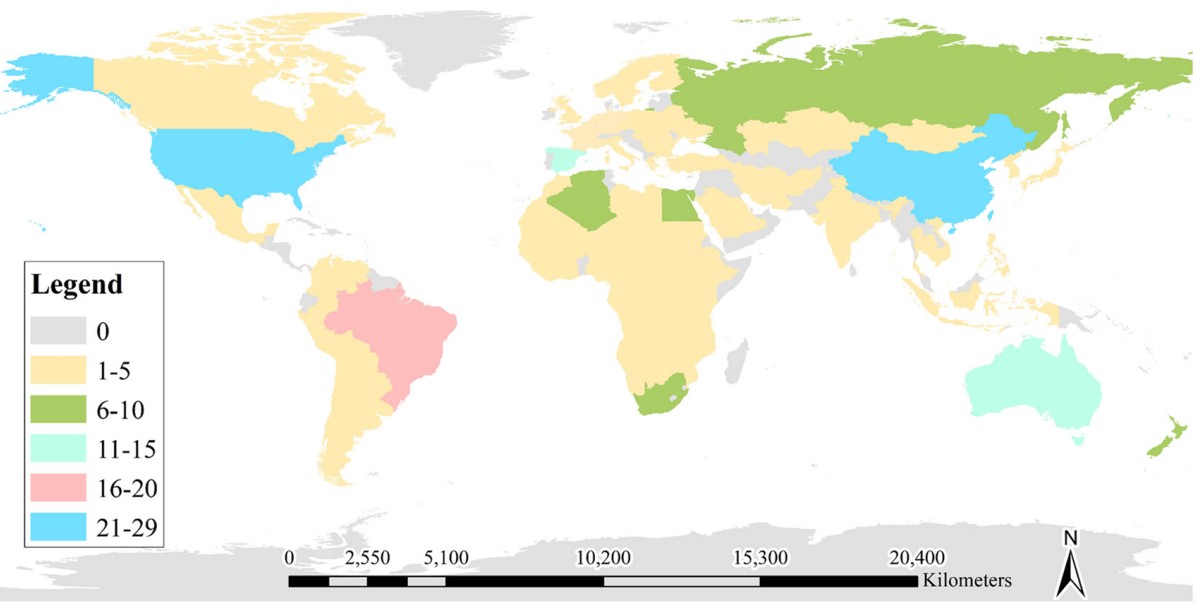

**Figure 4.** Spatial distribution and number of LAI studies conducted at country level.

### 3.3. Satellite Data Used

Advancements in remote sensing technology enabled researchers around the world to timely and accurately monitor vegetation. According to [22], the number of satellite systems for vegetation monitoring increased by 66% during 2016. Remote sensing enables the convenient collection of data dating back several years while providing reliable and accurate estimates of LAI and other biophysical attributes for different vegetation types.

The majority of forest, crop and grassland studies analysed in this paper used Landsat, followed by Sentinel and MODIS sensors. Freely accessible satellite data such as Landsat, Sentinel and MODIS gained popularity compared to commercial satellite data sources, such as World-View, IKONOS and QuickBird. Other commonly used sensors for estimating LAI were WorldView-3, SPOT, QuickBird and IKONOS. LiDAR data were mostly used on forests, followed by crops. There were only four studies on grasslands that used LiDAR data (Figure 5). In general, forests have a complex canopy structure; therefore, observing forest parameters using LiDAR was shown to be efficient [23], especially for reducing the impact of LAI saturation [24].

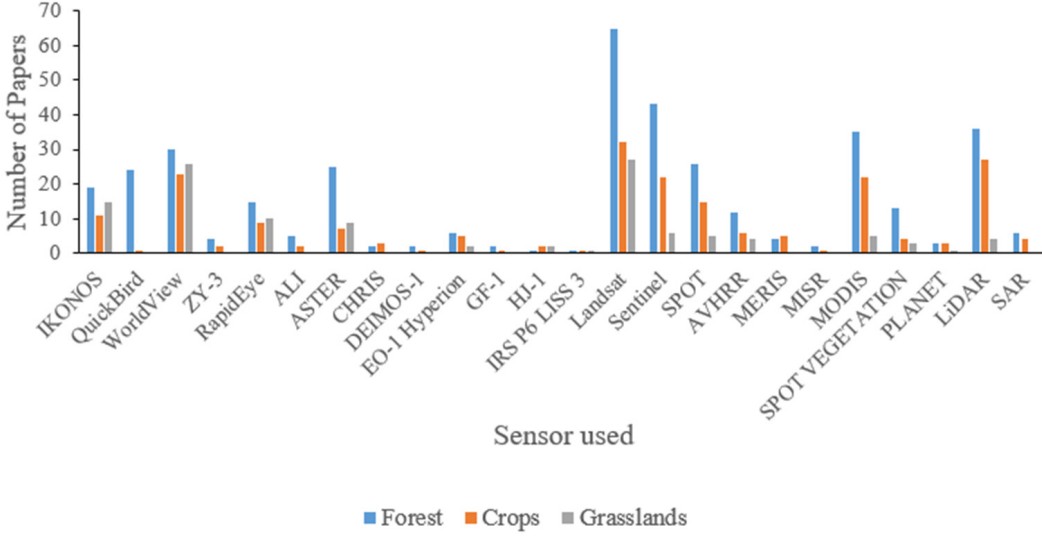

**Figure 5.** Sensors used to estimate LAI for forest, crop and grassland.

Passive sensors were the most widely used sensors for forest, crop and grassland LAI studies. Alternatively, active sensors such as radar systems and multisource remote sensing had a smaller proportion of studies. This review showed that the majority of forest and crop studies were carried out using active and passive sensor types, while grassland LAI studies mostly used passive sensors. There were no grassland LAI studies carried out using active remote sensing (Xu et al., 2020) [12]. The increase in freely available passive satellite data led to extensive use in forest, crop and grasslands. The number of LAI estimation studies using passive sensors was three times higher when compared to active remote sensing studies. The number of studies using unmanned aerial vehicles (AUV) increased since 2010, which enabled a quick turnaround time and real-time crop monitoring. There were a few forest and grassland LAI studies that used AUV compared to crop LAI studies. Figure 6 depicts the application of different sensor systems for forest, crop and grassland studies.

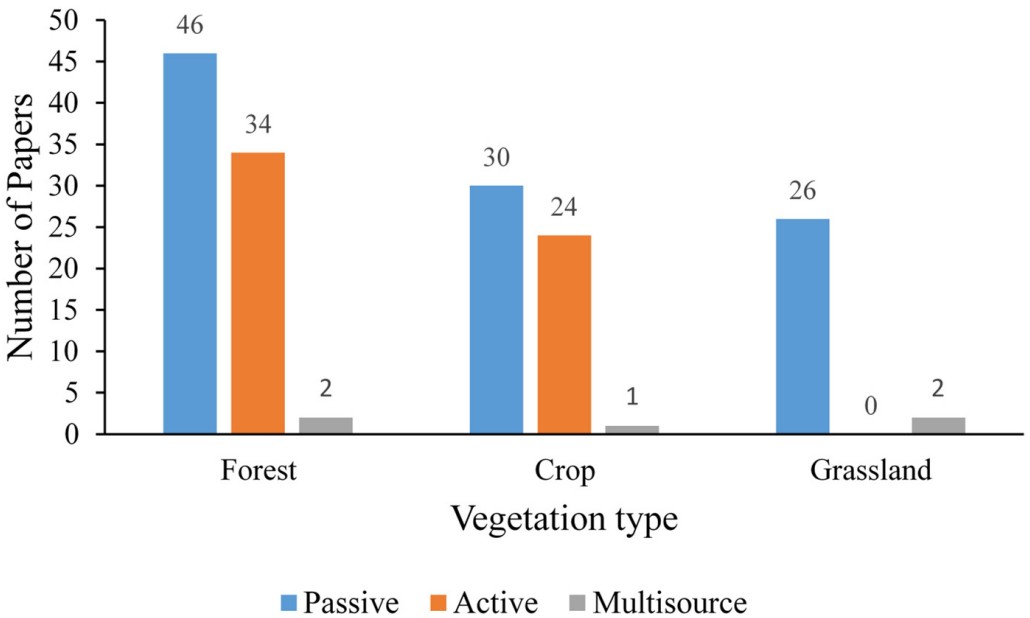

**Figure 6.** Application of different sensor systems for forest, crop and grassland.

In terms of commercial forests, according to Figure 5, most forest studies used passive sensors. LiDAR was the most used active remote sensing type for commercial forests. LiDAR was one of the most common active remote sensing systems, which detected canopy attributes by emitting light pulses to the target feature, measuring distance ranges and generating three-dimensional information about the canopy structure [25–27]. LiDAR was the most commonly used active remote sensing system for forestry [28,29], there were fewer crop studies and no studies for grasslands that used LiDAR.

### 3.4. LAI Estimation Models Used for Forest, Crop and Grassland Measures

Figure 7 shows the number of forests, crop and grassland LAI studies that used empirical, physical, hybrid and other models. The figure also shows the accuracy range of the methods used in the abovementioned models. To measure model performance, $R^2$ and RMSE were used. Of the 168 papers that were reviewed, 19 papers applied multiple models with a single data source. In addition, the review showed that few studies used image texture to estimate LAI and, therefore, this requires further investigation.

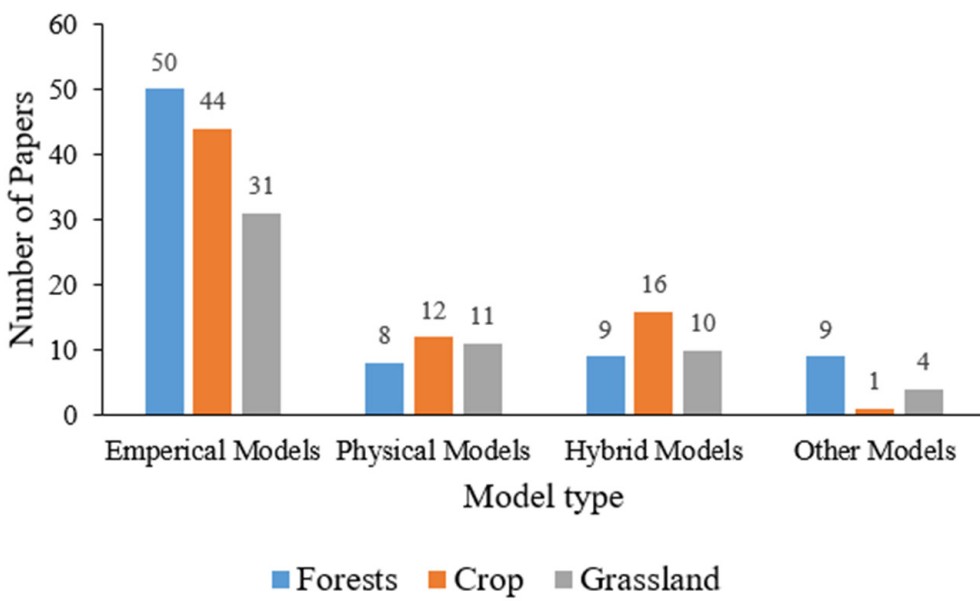

**Figure 7.** Distribution of different model types for forest, crop and grassland LAI estimations.

This review showed that empirical models were the most often used models for estimating the LAI of forest environment from 2010 to 2020. In terms of physical and hybrid models, forests had the lowest number of studies compared to crops and grasslands. The differences may be attributed to the typically complex forest structures. The number of crop studies that used physical and hybrid models, particularly PROSIAL, was higher compared to forests and grasslands. In terms of model types, vegetation index-based models and reflectance models were the most often applied model types for forests, crops and grasslands. Table 2 illustrates the models and sensors used for forest, crop and grassland studies and the level of accuracy achieved by each model and Table 3 shows the number of studies that applied each type of model. The number of applied models was higher than the number of studies reviewed because 19 papers used more than one model.

**Table 2.** Models and sensors used for forest, crop and grassland studies and the level of accuracy achieved by each model.

| | | Forest | | | Crop | | | Grassland | | | |
|---|---|---|---|---|---|---|---|---|---|---|---|
| Model | Sensors | N | R | RMSE | N | R | RMSE | N | R | RMSE | Method |
| Empirical Models | VIs based | 27 | 0.14–0.97 | 0.05–2.41 | 22 | 0.59–0.82 | 0.52–1.53 | 25 | 0.58–0.83 | 0.20–1.71 | Wide range of VIs |
| | Reflectance based | 15 | 0.58–0.97 | 0.10–1.08 | 9 | 0.45–0.95 | 0.03–1.82 | 10 | 0.36–0.95 | 0.02–1.83 | Regressions |
| | Derived metrics | 19 | 0.57–0.98 | 0.02–1.47 | 3 | 0.35–0.95 | 0.01–1.95 | 2 | 0.45–0.98 | 0.16–0.45 | Regressions |
| | Machine Learning | 8 | 0.84–0.95 | 0.43–1.95 | 13 | 0.58–0.97 | 0.34–1.94 | 9 | 0.44–0.97 | 0.14–1.66 | ML Algorithms |
| Physical Models | PROSAIL | 2 | 0.81–0.95 | 0.41–0.48 | 8 | 0.83–0.96 | 0.13–1.45 | 4 | 0.38–0.98 | 0.13–1.48 | |
| | DART | 2 | 0.75–0.87 | 0.46–0.52 | 0 | | | 0 | | | Look up tables (LUT) |
| | PROSPECT + DART | 1 | 0.77 | | 0 | | | 0 | | | |
| | 4-Scale bidirectional reflectance distribution (BRD) | 2 | 0.80–0.86 | 0.78–1.4 | 1 | 0.85 | 1.3 | 0 | | | Iterative optimization, LUTs |
| | Other models | 5 | 0.71–0.95 | 0.71–0.96 | 3 | 0.83–0.99 | 0.31–0.92 | 2 | 0.87–0.95 | 0.83–0.95 | LUTs, dynamic model, etc. |

**Table 2.** *Cont.*

| Model | Sensors | Forest | | | Crop | | | Grassland | | | Method |
|---|---|---|---|---|---|---|---|---|---|---|---|
| | | N | R | RMSE | N | R | RMSE | N | R | RMSE | |
| | PROSAIL | 5 | 0.63–0.92 | 0.26–1.12 | 9 | 0.58–0.96 | 0.3–1.16 | 5 | 0.67–0.98 | 0.27–1.13 | |
| | PARAS | 2 | 0.83–0.93 | 0.71–0.85 | 0 | | | 0 | | | |
| Hybrid Models | 4-Scale BRD | 2 | 0.43–0.68 | 0.31–1.08 | 1 | 0.86 | 0.43 | 0 | | | Regression and ML algorithms |
| | Other models | 5 | 0.21–0.99 | 0.20–1.86 | 1 | 0.89 | 0.5 | 1 | 0.78 | 0.64 | |
| Other models | | 6 | 0.41–0.096 | 0.01–0.92 | 3 | 0.43–0.97 | 0.42–0.98 | 1 | 0.67–0.099 | 0.01–1.94 | Regional phenology model, path length distribution model, etc. |

**Table 3.** Most used model types.

| Model | Number of Papers | Example |
|---|---|---|
| Reflectance based | 38 | Li et al. [30] |
| Vegetation index based | 47 | Qiao et al. [31] |
| Derived metrics | 29 | Zhang et al. [32] |
| Machine learning models | 11 | Lin et al. [33] |
| 4-Scale bidirectional reflectance distribution | 14 | Liu et al. [34] |
| PROSPECT + DART | 29 | Banskota et al. [35] |
| PROSAIL | 23 | Le Maire et al. [36] |
| Other models | 14 | Wang et al. [37] |

*3.5. Empirical-Based Models*

One of the commonly used remote sensing methods of deriving LAI from spectral data is the empirical-based approach [38]. In this approach, regression models are used to study the relationship that exists between the target variables; for example, LAI and its spectral reflectance. Empirical methods also use derived metrics and machine-learning methods to derive LAI. The number of studies exploring newly developed vegetation indices and models increased for forest, crop and grassland LAI estimations.

The $R^2$ of empirical models ranged from 0.14 to 0.99 and the RMSE ranged from 0.01 to 2.41 (Table 2). This indicates that the performance of empirical models was variable. Machine learning methods performed relatively consistently for forest in terms of the $R^2$ (0.85–0.93). Unlike forests, crops and grasslands had a highly variable performance in terms of $R^2$ (0.44–0.98). The use of machine-learning algorithms for estimating LAI showed positive results in a number of studies; examples include [32,33,39,40]. A study conducted by [39], using WorldView-2 imagery, demonstrated that machine-learning algorithms such as support vector machines (SVMs) and artificial neutral networks (ANN) can accurately predict LAI at tree species level. The SVM model which was based on validation data achieved an $R^2$ of 0.75. There was an overall increase in machine learning methods, especially for big data calculations.

There were more than 150 vegetation indices that were developed for mapping and estimating biomass, either using space borne, airborne and ground-based sensors [41]. Of the 150 vegetation indices, only a few were tested and validated [42]. Ref. [43] estimated LAI and other plant structural attributes using satellite images. This study used traditional vegetation indices such as the normalized difference vegetation index (NDVI) and found that NDVI had a strong correlation of $R^2 = 0.83$ with biomass and $R^2 = 0.70$ with LAI. Although different vegetation indices showed positive results for estimating LAI, there

was no vegetation index that is generic and can be applied consistently and continuously to estimate LAI regionally and globally.

### 3.6. Physical Models

Another method of deriving LAI from spectral data is through the inversion of radiative transfer (RT) or physical process models. Physical models are based on the propagation of light in plant canopies and are used to retrieve biophysical parameters of vegetation such as LAI. Most physical algorithms are rooted in the principles of reflectance model inversion and are operated through minimizing a merit function. Such a function yields a value for LAI by minimizing the summed differences between simulated and measured reflectances for all wavelengths [35].

Radiative transfer models (RTM) were shown in a number of studies as one of the successful methods of estimating LAI from remote sensing data [34–36,44]. The widely used and well validated RTMs included PROSPECT, which is a 1-Dimentional leaf reflectance model [45] and SAIL, which is a canopy reflectance model [46]. The performance of physical models in terms of $R^2$ (0.71–0.99) was relatively consistent compared to empirical models (0.14–0.99). The model with the highest number of applications in forest, crop and grassland studies was PROSPECT and SAIL radiative transfer models (PROSAIL), which could be attributed to its ease of use and robustness. From the studies that were reviewed in this paper, there were no crop and grassland LAI studies that applied the PROSPECT + discrete anisotropic radiative transfer model (DART), while four-scale bidirectional reflectance distribution (BRD) and DART models had the highest number of applications in forest environments.

Ref. [47] used the combined version of PROSPECT and SAIL known as PROSAIL to retrieve the LAI of saltmarsh from Sentinel-2 and RapidEye satellite data. This study achieved an $R^2$ of 0.59 and RMSE of 0.16 for Sentinel-2 and an $R^2$ of 0.65 and RMSE of 0.11 for RapidEye. The adaptability of this model was highly dependent on the type of remote sensing data, the ecosystem and the texture of the vegetation canopy. This model was not found to work well with coarse spatial resolution data; furthermore, it was more adaptable to homogeneous vegetation canopies.

Ref. [48] used the radiative transfer model to estimate the LAI of grasslands. The RTM-based method showed a higher accuracy of $R^2 = 0.64$ and RMSE = 42.67 gm$^{-2}$ when compared to other models. However, the disadvantages of the RTMs are that they are not as fast as other methods and they have many parameters which are difficult to acquire [48].

### 3.7. Hybrid Models

Hybrid models combine statistical and RTM input and output data to generate a simulation dataset which includes spectral data, vegetation indices and biophysical and biochemical properties of vegetation [49]. There are different hybrid model types, which include an iterative approach, optimisation method, look up tables (LUT) and artificial neutral networks. In terms of machine learning algorithms, ANN was most prominent type of hybrid model in the reviewed literature; this gained popularity because of its computational speed and retrieval performance [50]. ANN was reported in the literature as more accurate than empirical models for estimating LAI [51]. A number of studies applied hybrid models and achieved satisfactory results, and these include [40,42,52].

The $R^2$ range of hybrid models was 0.21–0.99 while the RMSE range was 0.26–1.13. The study by [53] compared a hybrid PROSAIL model and empirical model based on NDVI and simple ratio (SR). It found that PROSAIL performed better in terms of RMSE (0.38), while the RMSE of NDVI was 2.28 and 0.88 for SR. Hybrid models can be used to calibrate traditional vegetation indices such as NDVI, develop new vegetation indices and analyse the performance of vegetation indices.

*3.8. Other LAI Products*

There are a wide range of LAI products available. This section discusses some of the widely used LAI products that were operational after 2010. Advancements in radiative transfer models facilitated the development of operational global and regional LAI products. Although discrepancies between ground LAI data and digital LAI products still remain, improvements in atmospheric correction and radiometric calibrations improved LAI retrieval accuracy of many LAI products.

MODIS LAI is one of the most widely used LAI products which is based on the biome specific three-dimensional radiative transfer (3D RT) model. This model uses atmospherically corrected reflectance and a biome map to generate retrievals [36]. The MODIS LAI product is derived from the daily MODIS-Terra data and was available since 2000. Moreover, these data are available daily at 1 km spatial resolution.

There are many other LAI products that were developed. The multi-angle imaging spectrometer (MISR) is another LAI product that was used since 2000. This product is based on the same 3D RTM algorithms as the MODIS product, where the difference between MODIS and MISR is that MISR uses combined spectral data directional information collected by MISR. MISR algorithms provide LAI retrieval for 60–90% of the model input data [54].

GEOV2 LAI is another LAI product. This product is a Geoland2/BioPar project. GEOV2 is an improvement of the previous GEOV1 product in terms of continuity. GEOV2 data has less than 1% gaps compared to 20% of missing data/gaps in GEOV1 [55], GEOV2 also provides cleaner data that are less affected by radiometric and geometric noise. The data have been freely available since 2000 at the Copernicus portal (https://land.copernicus.eu/, accessed on 14 August 2020).

Global Land Surface Satellite (GLASS) LAI dataset was developed by Beijing Normal University and can be accessed at (http://www.bnu-datacenter.com, accessed on 14 August 2020). This product has a revisit time of eight days and operated from 1982 to 2012. From 2000 to 2012, the GLASS LAI product was derived using general regression neural networks (GRNNs) from MOD09A1, which is a MODIS land surface reflectance [56]. These data were available at 1 km for the globe.

GLOBMAP LAI is long term global LAI product which was developed in 1982 through a fusion of MODIS and historical advanced very high resolution radiometer (AVHRR) data. This product was developed by establishing a simple pixel-to-pixel relationship between MODIS and AVHRR datasets. These data were available at 8 km resolution.

EPS LAI is a global LAI product generated from AVHRR sensor on board the MetOp (Meteorological–Operational) satellite constellation. These data are available on a 10-day basis from https://landsaf.ipma.pt/en/, accessed on 14 August 2020. EPS LAI data are derived from an algorithm that uses the PROSAIL radiative transfer model as input and training data.

PROBA-V LAI is a first version Copernicus Global Land service global LAI product available at https://land.copernicus.eu/, accessed on 14 August 2020. This dataset is generated from Sentinel-3/OLCI, PROBAV sensors every five days and the spatial resolution of this dataset is 1/3 km.

The visible infrared imaging radiometer suite (VIIRS) LAI dataset is a National Aeronautics and Space Administration (NASA) product. This dataset was developed in 2018 and is still operational. The data can be downloaded at https://search.earthdata.nasa.gov/search, accessed on 14 August 2020. The data are available every eight days at a spatial resolution of 500 m. The VIIRS LAI dataset can be used to study energy absorption of broadleaves and coniferous vegetation canopies.

## 4. Discussion

This literature review investigated forms of LAI estimations using remote sensing technology for forests, crop and grasslands. The paper further unpacked trends in the use of different sensor systems, models, number of journal articles published and LAI

products. The review found that there were considerable advancements in terms of satellite system technology and models used for LAI and this led to improvements in LAI index estimation accuracy. Satellite systems and data improved in terms of spatial resolution, data accessibility, spectral resolution and satellite technology diversity. Satellites such as Pleiades Neo offer a spatial resolution of 30 cm, which is the best spatial resolution available today. New commercial and free satellite with higher spectral resolution entered the market, giving consumers a wide range of remote sensing technologies to choose from. The introduction of red edge band in most remote sensing technologies provided means for scientists to better understand vegetation attributes. Accurate LAI estimates of forests, crops and grassland are essential for improving the management and enhancement of the health of these vegetation types.

There was great progress in terms of satellite data availability and access since the launch of Landsat in 1972. The development of cloud satellite data access and processing platforms such as Google Earth Engine and Sentinel Hub enabled data scientists and researchers to access and process large data sets efficiently and quickly. Furthermore, these platforms were found to have built-in data modelling and analysis tools. Easy access to data meant the increase in the number of studies using free satellite data such as Landsat, Sentinel and lower spatial resolution moderate resolution imaging spectroradiometer (MODIS). Landsat is characterized by a spatial resolution of 30 m, and is the only existing satellite offering a multidecadal data for assessing, monitoring and mapping land changes. While Sentinel offers higher spatial resolution of 10 m and 13 spectral bands, it does not offer multidecadal data. MODIS is categorized as a coarse spatial resolution sensor as it offers a spatial resolution of 250 m to 1 km. Although MODIS provides multidecadal and consistent data, it cannot be used effectively to model ecosystems at local and regional scales [57]. The defining characteristics of commercial satellite include their high spatial and spectral resolution. WorldView-3 is one of the commonly used commercial satellite which was launched in 2014, making it the sixth Digital Globe's satellite in orbit (Satellite Imaging Corporation Houston, 2017). It joined satellites such as Ikonos which was launched in 1999, QuickBird in 2001, WorldView-1 in 2007, GeoEye in 2008 and WorldView-2 in 2009. Although commercial satellite data offer high spatial resolution and a better spectral and spatial resolution, they are still relatively costly. The most commonly used commercial satellite data in this review were SPOT, QuickBird, IKONOS and WorldView (Figure 5). The popularity of the aforementioned commercial satellite is because of high spatial resolution and higher revisit times. The results also showed an increasing trend in the fusion of data from different sensors and data with different spatial, spectral and temporal resolution.

Of the 168 forest, crop and grassland LAI estimation papers reviewed, the use of passive remote sensing in these studies was almost three times higher when compared to active remote sensing studies. In terms of active remote sensors, LiDAR was more widely used for forests than crops and grassland. This may be because discreet LiDAR data tended to have a poor penetration ability in short vegetation. These findings are consistent with the previous review paper conducted by [12], which reviewed research trends and future directions of LAI estimations using remote sensing from 1990 to 2020. Radar systems such as synthetic aperture radar (SAR) were mostly used for crops compared to forests and grasslands. Generally, crops are more dynamic throughout the growing season; therefore, radar systems that function under varying cloud conditions are useful for temporal or time-series crop analysis and can be used even during seasons of high cloud cover. Although grasslands are homogenous and less dynamic throughout the growing season, they usually cover larger areas, which poses a challenge when obtaining and processing data. The number of studies using unmanned aerial vehicles (AUV) increased since 2010, which enabled a quick turnaround time and real-time crop monitoring. This increase was especially notable in LAI estimation for crops. AUVs are equipped with micro sensors which can be used to obtain crop physiological and biochemical information at ultralow altitudes and low cost. Although satellite sensors can obtain crop image over a

large area, it can be difficult to obtain accurate biophysical information of crop canopy from a low spatial and temporal resolution satellite sensor.

Remote sensing studies on LAI for forests, crop and grasslands were conducted across a wide range of scales. Although the majority of studies were conducted at local and sub country scales (110), there were also studies conducted at global (7), continental (19) and country scales (29). There were also 14 studies that were conducted at variable scales. Analyzing the data of studies conducted at continental or global levels is challenging as it requires high computer processing power. Empirical, physical and hybrid model types have different advantages and limitations. Empirical models, as shown in the results section, were most commonly used models and were effective as they can be used across different data types to determine the relationship between LAI and spectral reflectance data. However, these methods utilized large statistical data inputs and can only be applied at certain locations, as they are highly dependent on vegetation types and canopy structural change [58]. Another limitation of empirical models is saturation problems. Although some studies showed that this problem can be mitigated by incorporating modified indices, this problem cannot be completely avoided when using optical imagery. According to [59], machine-learning algorithms offered a better way to analyze remote sensing data, as they do not make assumptions about variable distributions. However, it is important to understand the limitations of empirical methods such as machine-learning algorithms (e.g., multi-collinearity) before applying them.

Physical models, unlike empirical models, provide solutions developed without strong reliance on field data [47]. The great advantage of physical models is their applicability at wide range locations and scales, where they can also be applied to a wide range of vegetation types and canopy structures. The results showed that there was an increase in the application of all model types. However, in terms of physical models, PROSAIL was the most commonly used radiative transfer model. A challenge associated with physical models is that they are complicated, as they require many data inputs and are time-consuming to process.

Hybrid models are a combination of empirical models including machine-learning algorithms, regression methods and physical models. Ref. [47] used the combined version of PROSPECT and SAIL known as PROSAIL to retrieve the LAI of saltmarsh from Sentinel-2 and RapidEye satellite data. This study demonstrated that RTM was one of the best methods to estimate LAI from remotely sensed data. However, the adaptability of this model is highly dependent on the type of remote sensing data, ecosystem and the texture of the vegetation canopy. As mentioned in the results section, the current trend for forest, crop and grassland LAI estimation is the exploration of newly developed algorithms based on machine-learning methods, especially for big data calculation. However, there are still challenges associated with the use of hybrid models, which include the problem of instability and model overfitting. Problems associated with hybrid models are inherited from empirical models since hybrid models integrate the methods of empirical models.

There are other models which were less frequently reported in the literature. These models include the radiative transfer (GORT) model [60,61], geometric optical mutual shadowing (GOMS) model [62,63], geometric optical and a two-layer canopy-reflectance model (ACRM) [64,65], Kuusk–Nilson forest reflectance model [66,67] and discrete anisotropic radiative transfer (DART), which is a 3D canopy reflectance model developed by [68]. The above-mentioned models were developed to describe the effect of a three- and two-dimensional canopy structure on radiation, these models treated vegetation canopy as randomly distributed tree crowns which absorbed and scattered radiation passing through the crown. A review by [69] concluded that combining new algorithms and complementary information from various sensors lead to the development of better global LAI products. In terms of comparison of satellite LAI with model simulated LAI, different types of data validation and comparison methods were used. The most commonly used data comparison and validation method was a non-contact method that measured proportion of light penetrating through canopy using a plant canopy analyser known as Li-Cor LAI-2200;

this handheld device was found to be easy to use, less laborious and less time-consuming compared to destructive tree harvesting and non-destructive litter collection [70]. Other examples of data comparison and validation methods included the use of hybrid instruments such as multiband vegetation imager, which is an optical method that was used by [71] to validate LAI estimated from LiDAR data.

There are a number of global LAI products that were developed. For example, MODIS, MERIS, which are medium to coarse resolution data, are the most widely used satellite sensors to generate global LAI products. Ref. [9] reviewed the quality of MODIS (MOD15A2), Copernicus PROBA-V (GEOV1) and the recent EUMETSAT Polar System (EPS) LAI products for croplands LAI estimation. They found that the quality of LAI data from the above-mentioned satellite data were closely related to Sentinel-2 and Landsat 7/8 LAI data ($R^2$ = 0.90, RMSE = 0.50). Other popular LAI products include GEOV2 LAI, Global Land Surface Satellite (GLASS) LAI and GLOBMAP LAI.

The availability of satellite data which combine high revisit time with high spectral and spatial resolution fast tracked the development and validation of sophisticated models for LAI estimation. The current focus on improving the robustness of RTMs will lead to the development of even better and more accurate LAI products.

## 5. Conclusions

This paper reviewed 168 published articles on LAI estimations for forests, crops and grassland. The paper reviewed articles in terms of study sites, satellite sensors and statistical methods used to determine LAI. The results of the study indicate that there is an increasing trend in the use of all model types. However, the majority of the literature still focused on conventional statistically focused empirical models. There is also an increasing trend in the fusion of data from different sources. The fusion of multi-source data includes using data with different spatial, spectral and even temporal resolution. In terms of passive and active sensors, LiDAR demonstrated a great ability to detect forest canopy structures and estimate forest LAI even in mixed species plantations.

The development of high spatial resolution sensors such as WorldView-3 increased efficiency and improved results drastically. However, like other systems, remote sensing has its drawbacks. One of them is the dependency on weather conditions, as some measurements can only be taken under clear-skies conditions. Although a number of remote sensing methods of estimating LAI were developed, not a single method can be applied consistently and repeatedly for estimating LAI locally and continentally. The reason for this is variations in biophysical, environmental and topographic traits of vegetation in space and time. In the short term, empirical models that can be validated locally are recommended for forest and crop managers.

**Author Contributions:** Conceptualization, N.M. and R.L.; methodology, N.M. and R.L.; software, N.M. and R.L.; validation, N.M., R.L. and H.K.; formal analysis, N.M.; investigation, N.M.; resources, N.M., R.L. and H.K.; data curation, N.M.; writing—original draft preparation, N.M.; writing—review and editing, N.M., R.L. and H.K.; visualization, N.M.; supervision, R.L. and H.K.; project administration, N.M., R.L. and H.K. All authors have read and agreed to the published version of the manuscript.

**Funding:** This research received no external funding.

**Institutional Review Board Statement:** Not applicable.

**Informed Consent Statement:** Not applicable.

**Data Availability Statement:** Data sharing not applicable. No new data were created or analyzed in this study. Data sharing is not applicable to this article.

**Acknowledgments:** The authors of this paper acknowledge the support of the University of KwaZulu-Natal and the Mondi Group; which allowed for the successful completion of this research.

**Conflicts of Interest:** The authors declare no conflict of interest.

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
