# Peer review of "Forest, Crop and Grassland Leaf Area Index Estimation Using Remote Sensing: A Review of Current Research Methods, Sensors, Estimation Models and Accomplishments"

_applsci, doi:10.3390/app13064005_

Round 1

Reviewer 1 Report

The manuscript “Forest, crop and grassland Leaf Area Index estimation using remote sensing: A review of current research trends and accomplishments” is dealing with Forest, crop and grassland Leaf Area Index estimation by remote sensing model or sensor-based estimation. The aim of this paper is to review on remote sensing methods of estimating LAI across different forest ecosystems, crops and grasslands. As the authors mentioned that Leaf Area Index (LAI) is an important parameter in plant physiology and can be used to quantify foliage directly and as a measure of the photosynthetic active area and, thus, the area subject to transpiration in vegetation. The review of literature was based on published material between year 2010 to 2020. The manuscript gathered information on important aspect of remote sensing and will definitely attract the readers of Applied Sciences (ISSN 2076-3417) Journal.  Some suggestions are given below to improve the readability of MS.

·     Title

The title of a manuscript should be concise, precise and may represent all aspects of the study. Current title of the manuscript is representing the study.

·     Abstract

Abstract of the MS is looking fine to me but I guess more words than the journal requirement, so please follow journal requirements.

·     Introduction

Introduction is looking fine, but authors are requested to remove older citation mean before 2010, as you already mentioned in abstract a yard stick of review from 2010 to 2020. Please follow Journal format of citation etc.

·       M&M

Did you follow any statistically methods to analyze the citations?

·     Result and discussion

Result and discussion is also fine.

·     Conclusion

Conclusion is also fine.

Authors are requested to please follow the instruction to authors while reformatting the manuscript according to journal requirements.

All the best!   

Reviewer 2 Report

Review Summary:

It was my pleasure to read and review the paper entitled " Forest, crop, and grassland Leaf Area Index estimation using remote sensing: A review of current research trends and accomplishments” with a focus on reviewing remote sensing methods, platforms, and sensors on estimating Leaf Area Index of forests, crop, and grasslands for the period of 2010-2020. Authors have attempted to summarize findings in terms of popularity of satellite sensors and estimation models to conclude how successfully each techniques derive the leaf area index for various biomes. There is no doubt that this kind of review would be a valuable resource for future researcher to read merits and demerits of each approach in summarized form in a single research article for them to select an approach that best suits their LAI estimation goal.

I, however, have a few concerns that the authors need to address for this paper to be considered for publication.

I have outlined my comments below in two sections: Major comments and specific Comments.

Major comments:

1. I did not read the authors’ convincing points on why they chose this short time window (2010-2020). I also have questions about the number of articles reviewed in a review article like this. I am not entirely sure why, but Google Scholar results in more articles than those reported by authors for that period for the leaf area index estimation.

2. I recommend authors consider defining and discussing the major aspects of this review in elaborate detail. An elaborate review of those main components (sensors, models) may serve as a single scholarly resource if authors have a section outlining what are the defining characteristics of the three landcover types: forest, crop, and grassland; various remote sensing sensors they touched upon (Landsat, MODIS, WorldView-3, UAVs, etc.), and the models (empirical, physical, hybrid, and other models).

3. The review presented the statistics of agreement primarily in a range of values. I would have liked to see those values listed by most used models or statistical methods.  

4. Also, I did not note much discussion on what those LAI estimations were compared against. The authors have noted down some VI-based LAI estimations but have not summarized what kind of ground-reference data was used to compare the models’ performance and how were those collected.  

5. Discussions: In the discussion, the authors noted that there have been considerable advancements in terms of satellite technology and models. Because there was not much of that exemplified (advancement in what aspects? – resolution, platform height, sensors’ sensitivity of the specific remote sensors, and what aspects of models) written down in the prior section, it appears as a vague statement derived from other literature rather than from analyzed summary.

The discussion did a poor job by not focusing to discuss the main outcomes of the reviews. This section, most of the time, repeats the results or adds new information from the literature rather than trying to rationalize in detail why the specific satellite sensors, models, and platforms were more (or less) preferred than the others in estimating LAIs across different vegetated areas.

Specific Comments:

Line 15: Correct the typo for the word: estimable  

Line 117: Number of the articles reported on this line (165) does not agree with the number used subsequently. On this line, it says the dataset was reduced to 165 articles, whereas the results section starts with a sentence saying 168 articles.

Lines 214,226, 245: Example occasion where authors have used the acronym VI without spelling it out for the first time.

Line 260: Another occasion where I saw the acronym DART without spelling it out and without defining it.

There are other instances too (e.g., PROSAIL, BRD….)

Line 264: Do those two different R2 values correspond to two different sensors listed? That is not made clear in writing.

Line 265-266: Need more elaborate rationale on why authors said “adaptability is dependent on the sensor, ecosystem, vegetation characteristics".

 Line 356-357: Because the proper definition of satellite system was not considered, it appears as if I am reading the “radar system” here for the first time in this paper. 
